# Waning of SARS-CoV-2 booster viral-load reduction effectiveness

Matan Levine-Tiefenbrun [1,2], Idan Yelin[1], Hillel Alapi[3], Esma Herzel[3], Jacob Kuint[2,3], Gabriel Chodick [2,3], Sivan Gazit [3], Tal Patalon [3✉] & Roy Kishony [1,4✉]

The BNT162b2 COVID-19 vaccine has been shown to reduce viral load of breakthrough infections (BTIs), an important factor affecting infectiousness. This viral-load protective effect has been waning with time post the second vaccine and later restored with a booster shot. It is currently unclear though for how long this regained effectiveness lasts. Analyzing Ct values of SARS-CoV-2 qRT-PCR tests of over 22,000 infections during a Delta-variant-dominant period in Israel, we find that this viral-load reduction effectiveness significantly declines within months post the booster dose. Adjusting for age, sex and calendric date, Ct values of RdRp gene initially increases by 2.7 [CI: 2.3-3.0] relative to unvaccinated in the first month post the booster dose, yet then decays to a difference of 1.3 [CI: 0.7-1.9] in the second month and becomes small and insignificant in the third to fourth months. The rate and magnitude of this post-booster decline in viral-load reduction effectiveness mirror those observed post the second vaccine. These results suggest rapid waning of the booster's effectiveness in reducing infectiousness, possibly affecting community-level spread of the virus.

[1] Faculty of Biology, Technion - Israel Institute of Technology, Haifa, Israel. [2] Sackler Faculty of Medicine, Tel-Aviv University, Tel-Aviv, Israel. [3] Maccabitech, Maccabi Health Services, Tel Aviv, Israel. [4] Faculty of Computer Science, Technion - Israel Institute of Technology, Haifa, Israel. ✉email: patalon_t@mac.org.il; rkishony@technion.ac.il

The Pfizer/BioNTech BNT162b2 vaccine[1,2] demonstrated high real-life effectiveness in prevention of infection and disease[3–5] as well as in reducing viral loads of break-through infections (BTIs)[6]. Indeed, national vaccination campaigns were initially followed by reduced infection rates, with the effect extending beyond the individual vaccinees to the community level[7–9]. Considering longer post-vaccination times, studies have shown that while vaccine effectiveness in preventing severe disease declines only mildly[10], the effectiveness of protection against infection[2,11–13] and of reducing viral loads of BTIs strongly declines within several months[14]. A booster (third) dose was associated with restoring effectiveness of prevention of infection and disease[15], as well as with reduction of BTI viral-loads[14]. Yet, it is unknown for how long these booster's regained effectiveness lasts and whether there will be a need for additional booster shots in the future[16]. In this work we show that this effectiveness declines within months after the booster dose, mirroring the decay after the second dose.

## Results

Here, we focus on the booster's association with reduced viral loads over time. We retrospectively collected and analyzed the reverse transcription quantitative polymerase chain reaction (RT–qPCR) test measurements of three SARS-CoV-2 genes—E, N and RdRp (Allplex 2019-nCoV assay, Seegene)—from positive tests of patients of Maccabi Healthcare Services (MHS). We focus on infections of adults over the age of 20 between June 28 and November 29 2021, before the Omicron variant started circulating in Israel, and when Delta was the dominant variant in the country (over 93%)[17]. Crossing this dataset with vaccination data, we identified in total 5229 infections of unvaccinated, 16,038 BTIs of 2-dose-vaccinated and 1390 BTIs of booster-vaccinated individuals (Methods: "Vaccination status", Supplementary Table 1).

For each of the three viral genes, we built a multivariable linear regression model for the relationship between Ct value and vaccination at different post-second-vaccination time bins and post-booster time bins, adjusted for sex, age and calendric date (Methods; $n = 22,657$ infections). Consistent with our previous reports[6,14], regression coefficients for second-dose vaccinated, compared to unvaccinated, started with 4.2 [95% CI: 2.2–6.1] for BTIs occurring 7–30 days post 2nd vaccine dose and then decayed over time, effectively vanishing 6 months or longer post vaccination (RdRp gene, Fig. 1; Supplementary Fig. 1 for the E and N genes). Then, as previously observed[14], the booster restored effectiveness, with the Ct increasing by 2.7 [CI: 2.3–3.0] for infections occurring 7–30 days post the booster, corresponding to more than 6-fold reduction in viral load (RdRp gene, Fig. 1, Methods). However, this booster-associated viral-load reducing effectiveness rapidly declined, reaching 1.3 [CI: 0.7–1.9] and 0.8 [CI: −0.1–1.8] for BTIs occurring 31–60 and 61–120 days post the booster shot, respectively. Of note, age distributions of individuals in the above 3 post-booster time bins were very similar (Supplementary Fig. 2). A time-differential model further quantified these reductions and their statistical significance, yielding a difference of 2.6 [CI: 0.1–5.0, $P = 0.04$] and 1.4 [CI: 0.7–2.0, $P = 2.5 \times 10^{-50}$] between the Ct's of infections in the first and second time bins (7–30 versus 31–60 days) post the second dose and post the booster, respectively (Methods). These differences correspond to similar within-month percent decreases of 62% (2.6/4.2) and 52% (1.4/2.7) in Ct-reduction post the second dose and the booster shot, respectively. Modeling the time dependency as an exponential decline showed mild but statistically insignificant extension of the decay times post the booster (decay coefficients: 48 [CI: 30–73] days after the second dose and 65 [CI: 43–110] days after the booster, Methods). Similar trends were also observed for the 2 other genes, N and E (Supplementary Fig. 1). To avoid effects

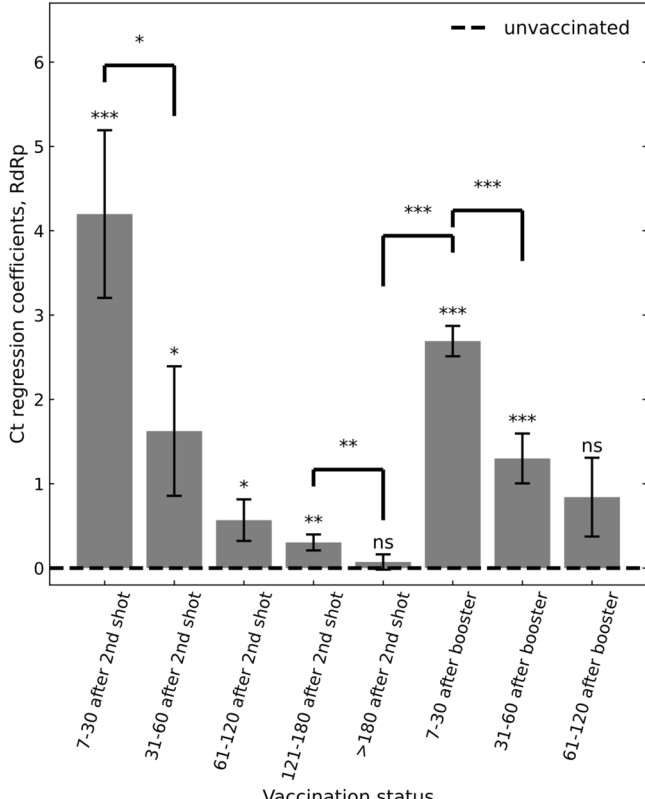

**Fig. 1 Association of infection Ct with two-dose vaccination and the booster.** Ct regression coefficients, indicating an infection Ct relative to unvaccinated control group (dashed line), show an initial increase in Ct in the first month after the second vaccination dose, which then gradually diminishes. Increased Ct is restored after the booster, yet this regained effectiveness also gradually diminishes. Coefficients were obtained by multivariate linear regression analysis, adjusting for age, sex and calendric date (Methods; $n = 22,657$). Error bars represent one standard error of the mean. All P values are two sided: 'ns', not significant, *$P < 0.05$, **$P < 0.01$, ***$P < 0.001$. Data are shown for Ct of the RdRp gene; for genes N and E, see Supplementary Fig. 1a, b.

of severe cases on the results, we also analyzed the data again while excluding hospitalized patients, obtaining similar results (Supplementary Fig. 3).

## Discussion

Our study has a number of limitations. First, although viral load is a common proxy for infectiousness, a correlation between viral loads and infectability is not fully established. Second, differences in health-seeking behaviors may affect the timing at which individuals are tested and as viral load is associated with time post infection, such differences may in principle bias our results. Yet, we note that we minimized this effect by considering only the first positive test for each patient, which is typically performed within 5–6 days following infection, namely 1–2 days following symptoms[15,18–20]. We further note that quantification of waning is based on differences between Ct's among vaccinated patients, presumably canceling some of the health-seeking biases. Lastly, the waning of immunity is observed during a Delta dominant period and additional data is needed to understand how it impacts other common variants such as the Omicron.

In total, these initial observations show that the BNT162b2 booster regained effectiveness in reducing BTI viral loads is declining within months, in a rate and magnitude mirroring those observed post the second vaccine. Pending post-booster

time-dependency data also on the effectiveness of disease prevention, these results warrant consideration of the long-term effectiveness of the mRNA vaccine and the possible benefit and anticipated duration of additional booster shots.

## Methods

**Data collection.** Anonymized SARS-CoV-2 RT–qPCR Ct values were retrieved for all of the positive samples taken between 28 June and 29 November 2021, and tested at the MHS central laboratory. Patient data included sex, year of birth, any record of COVID-assocaited hospitalization and a baseline condition of immunosuppression. For patients with multiple positive tests, only the first positive test was included (to minimize tests carried long after infection). Vaccination dates for these patients were retrieved from the centralized database of MHS. Immunocompromised patients were excluded (974). Patients were excluded if their first positive sample was between the first shot and one week after the second shot (where the vaccine is presumably not yet effective). Patients with a first positive sample within the first week following the booster shot were also excluded for the same reason. Additionally, patients with positive test results before the study period were excluded. For each test, Ct values for *E* gene, *RdRp* gene, *N* gene and the internal control were determined using Seegene proprietary software for the Allplex 2019-nCoV assay (CFX Manager Software V3.1) after the standard oronasopharyngeal swab specimen collection procedure. The same PCR model was used for all of the tests (Bio-Rad CFX96 Real-Time PCR Detection System).

**Vaccination status.** Patients tested positive 7 days or more after the second shot were regarded as 'vaccinated'. Patients tested positive 7 days or more after the third shot were regarded as 'booster-vaccinated'. Patients tested positive less than 7 days after the second shot were excluded, as well as patients tested positive between 0 and 7 days after their booster shot.

**Linear regression.** For each of the three viral genes, we calculated the linear regression of raw untransformed Ct values as a function of time since second shot (length-8 onehot vector indicating 0/1 for time bins of 7–30 days, 31–60 days, 61–120 days, 121–180 days, >180 days after the second shot; 7–30 days, 31–60 days, 61–120 days after the booster; all 0's for unvaccinated), sex (0/1, female/male), age (bins of 10 years), calendric date (number of days since 28 June, 2021) and a quadratic variable of calendric date. Models were implemented using Python's statsmodels library, version 0.9.0 and scipy library, version 1.1.0.

**Change in Ct over time.** To calculate the change in Ct between post-vaccination time bins, we applied the same linear regression model as above, except replacing the length-8 onehot vector of post-vaccination time with a 8-length time-accumulated binary vector indicating [0,0,0,0,0,0,0,0], [1,0,0,0,0,0,0,0], [1,1,0,0,0,0,0,0], [1,1,1,0,0,0,0,0], [1,1,1,1,0,0,0,0], [1,1,1,1,1,0,0,0], [1,1,1,1,1,1,0,0], [1,1,1,1,1,1,1,0], [1,1,1,1,1,1,1,1] for infections in unvaccinated, or 7–30 days, 31–60 days, 61–120 days, 121–180 days and over 180 days post the second vaccination, or 7–30 days, 31–60 days and 61–120 days post the booster, respectively.

**Log transformation of Ct values.** We approximate 1 Ct unit to a factor of 2 in the number of viral particles per sample: $viral\ load\ reduction\ factor = 2^{\Delta Ct}$ or $\Delta Ct = \log_2$ (viral load reduction factor).

**Exponential decline modeling.** To quantify decline rates, we fitted to the Ct data a model where time post vaccination is modeled as an exponential decline:

$$Ct = \eta_1 A_1 e^{-t_1/\tau_1} + \eta_2 A_2 e^{-t_2/\tau_2} + linear\_components + const,$$

where, $\eta_1$ is a binary variable indicating whether a person is post second dose; $\eta_2$ is a binary variables indicating whether a person is post third dose; $t_1$, $t_2$ are times post the second dose and the booster, respectively; $A_1$ and $A_2$ are fitted parameters indicating an initial increase in Ct post the second does and the booster, respectively; $\tau_1$, $\tau_2$ are fitted parameters indicating the decay times post the second does and the booster, respectively. The *linear_components* term includes contributions of sex, age group, calendric time and calendric time squared as decribed in the Linear Regression model above.

In the text, we report the decay times $\tau_1$, $\tau_2$. Statistical confidence intervals were quantified by bootstrapping (5000 re-sampling simulations) with restricting bounds of ±1.5 from the nominal result.

**Reporting summary.** Further information on research design is available in the Nature Research Reporting Summary linked to this article.

## Data availability

According to Israel Ministry of Health regulations, individual-level data cannot be shared openly. Specific requests for remote access to de-identified data should be referred to KSM, the Maccabi Healthcare Services Research and Innovation Center. Requests sent to Sarit Chehanowitz (chehanow_s@mac.org.il) will be considered within 21 d pending IRB approval and Maccabi Healthcare Services regulations. Source data are provided with this paper.

## Code availability

The code for the analysis is available at https://github.com/matanlevine/Waning-of-SARS-CoV-2-booster-viral-load-reduction-effectiveness.

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

## Acknowledgements

This work was supported by the Israel Science Foundation KillCorona-Curbing Coronavirus Research Program (grant no. 3633/19 to R.K. and G.C.) and a fellowship from the Edmond J. Safra Center for Bioinformatics at Tel-Aviv University (to M.L.-T.). The funders had no role in study design, data collection and analysis, decision to publish or preparation of the manuscript.

## Author contributions

Study design: M.L.-T., I.Y., T.P., S.G., G.C., and R.K. Data collection: M.L.-T., I.Y., E.H., and H.A. Data analysis: M.L.-T., I.Y., and R.K. Data interpretation: M.L.-T., I.Y., J.K., T.P., S.G., G.C., and R.K. Writing: M.L.-T., I.Y., and R.K, with comments from all authors.

## Competing interests

The authors declare no competing interests.

## Ethical approval

This study was approved by the MHS (Maccabi Healthcare Services) Institutional Review Board (IRB). Due to the retrospective design of the study, informed consent was waived by the IRB, and all identifying details of the participants were removed before computational analysis.
