## [Peer Review File · Nature Communications]

Waning of SARS-CoV-2 booster viral-load reduction effectivenessReviewers' Comments:

Reviewer #1:

Remarks to the Author:

The authors presented a unique and interesting study about the temporal reduction of Ct values among infected individuals who received the third dose of vaccine (or booster) in Israel. The manuscript is well written, and the methods are clear.

While I do think the data are important, I am not quite sure if this manuscript has added sufficient novel and important insights into the COVID-19 research. Considering the length of the manuscript, it would probably be more appropriate to reformat it as a research letter. Please see below for my specific comments.

1. Is it possible to compare the viral load reduction over time after the second dose as well? It would be interesting to see the comparison between the second and third dose. Also, as mentioned by the authors, there is not an established correlation between Ct value and vaccine effectiveness against transmissibility yet. Adding data of Ct value after the second dose of vaccine would provide more insights to this topic.

2. Viral load or Ct value also differs over time during the time course of illness. It is not clear to me whether the authors have also included the variable "time since infection" or "time since symptom onset" for breakthrough infections included in the analysis. I suggest adding more details about the analysis.

3. Following up the comment above, more details about methods should be included, e.g. (log-) transformation of Ct values, the regression model, model selection, and model fit etc.

4. In view of the emergence of omicron, how would this model be revised to incorporate the effects of increased transmissibility, reduced susceptibility from past infection, vaccination, and immune escapes in the context of omicron?

Reviewer #2:

Remarks to the Author:

Studies have shown that while vaccines are effectiveness in preventing severe disease over time, the effectiveness of protection against infection and of reducing viral loads of BTIs strongly declines within several months. The manuscript entitled "Waning of SARS-CoV-2 booster viral-load reduction effectiveness" by Levine-Tiefenbrun describes research through the utilization and systematic statistical analysis of retrospectively collected laboratory results from COVID-19 cases in Israel. It was previously established in cohorts of more than 20 thousand cases that Pfizer/BioNTech BNT162b2 vaccine effectiveness in reducing SCV2 viral-load significantly declines within months post the booster dose. In the first month post booster dose, Ct values initially increased significantly relative to unvaccinated; yet then declined to small or insignificant in the third to fourth months post booster; indicating rapid waning of the booster's effectiveness in reducing infectiousness.

The authors focus on the booster's association with reduced viral loads over time by retrospectively collecting and analyzing RT-qPCR CT values of SARS-CoV-2 genes E, N and RdRp from > 20 Y/O between June and November, 2021 (when Deelta was the dominant variant and before Omicron transmission) in Israel. 5,229 infections of unvaccinated, 16,038 BTIs of 2-dose-vaccinated and 1,390 BTIs of booster-vaccinated individuals were included in the study. Consistent with previous reports, reduction of viral load effectiveness for second dose vaccinated, compared to unvaccinated, decayed over time and vanished by 6 months. Then, the booster restored effectiveness with a 6-fold reduction in viral load. However, this booster-associated viral-load reducing effectiveness rapidly declined within the following 4 months post the booster shot. These differences correspond to within-month percent

decreases of 62% and 52% in Ct-reduction post the second dose and the booster shot, respectively. Exclusion of hospitalized cases rendered similar results. These evaluations indicate that the BNT162b2 booster regains effectiveness in reducing viral loads in BTI of the Omicron variant; but this effectiveness declines within months of the second dose or the booster, in a rate and magnitude mirroring those observed post the second vaccine previously observed for the Delta variant. Post-booster time-dependency on the effectiveness of disease prevention is important to establish as SCV2 variants continue to appear and rapidly spread in vaccinated populations. The knowledge gained from this type of studies inform public health programs on the long-term effectiveness and benefits of mRNA vaccines and the optimal time intervals between additional booster shots. Though these studies may mirror others in the rapidly evolving landscape of peer-reviewed publications on COVID-19; this reviewer finds the analysis to be thorough and relevant for public health action.

Reviewer #1 (Remarks to the Author):

The authors presented a unique and interesting study about the temporal reduction of Ct values among infected individuals who received the third dose of vaccine (or booster) in Israel. The manuscript is well written, and the methods are clear.

We thank the reviewer for their thorough and supportive assessment of the study.

While I do think the data are important, I am not quite sure if this manuscript has added sufficient novel and important insights into the COVID-19 research. Considering the length of the manuscript, it would probably be more appropriate to reformat it as a research letter. Please see below for my specific comments.

1. Is it possible to compare the viral load reduction over time after the second dose as well? It would be interesting to see the comparison between the second and third dose. Also, as mentioned by the authors, there is not an established correlation between Ct value and vaccine effectiveness against transmissibility yet. Adding data of Ct value after the second dose of vaccine would provide more insights to this topic.

We thank the reviewer for their comment. We have now rephrased the results and the methods paragraphs to more directly indicate the similarity of the decline in effectiveness observed after the second dose compared to the decline observed after the third dose (with reference to Fig 1). Furthermore, following the referee's comment, we added new analysis modeling the Ct over time with an exponential decline, thereby quantifying the decay rates post the booster and the post the second dose. These results showed only mildly longer, and not statistically significant, decay rate post the booster compared to post the second dose. These viral load reduction over time after the second dose as well as after the third dose are provided in the text.

2. Viral load or Ct value also differs over time during the time course of illness. It is not clear to me whether the authors have also included the variable "time since infection" or "time since symptom onset" for breakthrough infections included in the analysis. I suggest adding more details about the analysis.

We agree with the reviewer that Ct values differ over time since disease onset. However, since there are no records of disease onset date, it is not included in our model. To minimize the potential effect of time since disease onset we only consider in our model the first positive test result for each patient (see revised Methods: Data collection). We added a sentence indicating this limitation, while pointing out that, since we only report *differences* in Ct values among patients, such potential biases should mostly cancel out (unless the tendency to get tested early versus late post infection is itself a function of the time from immunization, which seems less likely).

3. Following up the comment above, more details about methods should be included, e.g. (log-) transformation of Ct values, the regression model, model selection, and model fit etc.

Following the reviewer's comment we have now added Methods: Log transformation of Ct values. We also clarify in the methods that the model fits the raw un-transformed Ct values (which by nature of the PCR methodology are themselves a logarithmic proxy of the viral load). Finally, we verified that the model choice and all specific features are listed under *Methods: Linear regression*.

4. In view of the emergence of omicron, how would this model be revised to incorporate the effects of increased transmissibility, reduced susceptibility from past infection, vaccination, and immune escapes in the context of omicron?

As the reviewer correctly comments, our study is focused on a Delta-dominant period in Israel. Following this comment we have now revised our discussion to point out potential differences in the model once applied to future data from a period when either the Omicron, or any other future strain, is dominant. We also now emphasize that the observed decline reflects the immune response to the vaccine and as long as we are still using the same vaccines, we anticipate similar waning time scales.

Reviewer #2 (Remarks to the Author):

Studies have shown that while vaccines are effectiveness in preventing severe disease over time, the effectiveness of protection against infection and of reducing viral loads of BTIs strongly declines within several months. The manuscript entitled "Waning of SARS-CoV-2 booster viral-load reduction effectiveness" by Levine-Tiefenbrun describes research through the utilization and systematic statistical analysis of retrospectively collected laboratory results from COVID-19 cases in Israel. It was previously established in cohorts of more than 20 thousand cases that Pfizer/BioNTech BNT162b2 vaccine effectiveness in reducing SCV2 viral-load significantly declines within months post the booster dose. In the first month post booster dose, Ct values initially increased significantly relative to unvaccinated; yet then declined to small or insignificant in the third to fourth months post booster; indicating rapid waning of the booster's effectiveness in reducing infectiousness. The authors focus on the booster's association with reduced viral loads over time by retrospectively collecting and analyzing RT-qPCR CT values of SARS-CoV-2 genes E, N and RdRp from > 20 Y/O between June and November, 2021 (when Deelta was the dominant variant and before Omicron transmission) in Israel. 5,229 infections of unvaccinated, 16,038 BTIs of 2-dose-vaccinated and 1,390 BTIs of booster-vaccinated individuals were included in the study. Consistent with previous reports, reduction of viral load effectiveness for second dose vaccinated, compared to unvaccinated, decayed over time and vanished by 6 months. Then, the booster

restored effectiveness with a 6-fold reduction in viral load. However, this booster-associated viral-load reducing effectiveness rapidly declined within the following 4 months post the booster shot. These differences correspond to within-month percent decreases of 62% and 52% in Ct-reduction post the second dose and the booster shot, respectively.

Exclusion of hospitalized cases rendered similar results. These evaluations indicate that the BNT162b2 booster regains effectiveness in reducing viral loads in BTI of the Omicron variant; but this effectiveness declines within months of the second dose or the booster, in a rate and magnitude mirroring those observed post the second vaccine previously observed for the Delta variant. Post-booster time-dependency on the effectiveness of disease prevention is important to establish as SCV2 variants continue to appear and rapidly spread in vaccinated populations. The knowledge gained from this type of studies inform public health programs on the long-term effectiveness and benefits of mRNA vaccines and the optimal time intervals between additional booster shots. Though these studies may mirror others in the rapidly evolving landscape of peer-reviewed publications on COVID-19; this reviewer finds the analysis to be thorough and relevant for public health action.

We thank the reviewer for their thorough review and their positive support.

Reviewers' Comments:

Reviewer #1:

Remarks to the Author:

Thank you for responding to my earlier comments. I have no other questions or comments above the study.